# The Down-Shifting Luminescence of Rare-Earth Nanoparticles for Multimodal Imaging and Photothermal Therapy of Breast Cancer

**DOI:** 10.3390/biology13030156

**Published:** 2024-02-28

**Authors:** Tingting Gao, Siqi Gao, Yaling Li, Ruijing Zhang, Honglin Dong

**Affiliations:** The Second Clinical Medical College, Shanxi Medical University, Taiyuan 030001, China; gaotingting@sxmu.edu.cn (T.G.); gaosiqi2@sxmu.edu.cn (S.G.);

**Keywords:** NIR-II, breast cancer, photothermal therapy, multimodal imaging

## Abstract

**Simple Summary:**

The global cancer report for 2020 highlights a noteworthy surge in new cases of breast cancer, exceeding lung cancer incidence for the first time and emerging as the foremost global cancer manifestation. Breast cancer has become a prominent contributor to female cancer-related mortality. Surgical removal continues to be a primary therapeutic recourse for breast cancer patients, aiming to extend survival by addressing residual tumor foci during surgery. However, the challenge lies in achieving complete excision due to the ambiguous boundaries of tumors. NIR-II imaging, known for its heightened sensitivity, superior spatiotemporal resolution, and enhanced optical penetration depth compared to conventional optical imaging, stands out as an optimal choice for preoperative and real-time intraoperative imaging, particularly in live fluorescence imaging. Within this study, rare-earth nanoparticles α-Er NPs exhibit remarkable imaging quality in breast cancer, incorporating X-ray computed tomography and ultrasound, collectively enhancing precision in cancer diagnosis and treatment. Additionally, the study explores the photothermal therapeutic effects of α-Er NPs. Through the combination of NIR-II imaging technology with rare-earth probes, the research pioneers a secure and effective novel strategy for the precise diagnosis and treatment integration of breast cancer, establishing a robust foundation for future advancements in medical applications.

**Abstract:**

Nanotheranostic agents capable of simultaneously enabling real-time tracking and precise treatment at tumor sites play an increasingly pivotal role in the field of medicine. In this article, we report a novel near-infrared-II window (NIR-II) emitting downconversion rare-earth nanoparticles (RENPs) to improve image-guided therapy for breast cancer. The developed α-NaErF4@NaYF4 nanoparticles (α-Er NPs) have a diameter of approximately 24.1 nm and exhibit superior biocompatibility and negligible toxicity. RENPs exhibit superior imaging quality and photothermal conversion efficiency in the NIR-II range compared to clinically approved indocyanine green (ICG). Under 808 nm laser irradiation, the α-Er NPs achieve significant tumor imaging performance and photothermal effects in vivo in a mouse model of breast cancer. Simultaneously, it combines X-ray computed tomography (CT) and ultrasound (US) tri-modal imaging to guide therapy for cancer. The integration of NIR-II imaging technology and RENPs establishes a promising foundation for future medical applications.

## 1. Introduction

Breast cancer is a prominent cause of cancer incidence, disability, and mortality in women, impacting one in every twenty individuals worldwide and as many as one in eight in high-income countries. Notably, the incidence of breast cancer is still on the rise [1,2]. So far, mammography, magnetic resonance imaging (MRI), ultrasound imaging, and CT have become normally applied to diagnose breast cancer. However, these conventional techniques are limited in sensitivity and specificity, and most of them are hardly available in surgery. Consequently, precise detection of breast cancer during surgery remains a great challenge [3,4].

In the past decades, intraoperative fluorescence molecular imaging (FMI) has revolutionized the way to resect malignancies with remarkable imaging speed, sensitivity, and resolution. Nowadays, FMI has become a valuable tool for improving tumor staging diagnosis, monitoring treatment response, and detecting recurrence or residual diseases [5,6]. Mostly, FMI is performed in the first near-infrared window (NIR-I, 700~900 nm). Although NIR-I has better performance than visible light, the autofluorescence effects and limited perpetration depth still hinder its further application [7,8].

Due to its high sensitivity, superior spatiotemporal resolution, and enhanced penetration depth compared to traditional optical methods, NIR-II imaging shows promise for in vivo fluorescence imaging, making it an ideal candidate for preoperative and intraoperative real-time imaging [9]. Therefore, the development of NIR-II fluorescence probes with high luminescence efficiency is crucial for enhancing the quality of tumor NIR-II fluorescence imaging.

With the advancement of imaging utilizing the second near-infrared window (NIR-II, 1000~1700 nm), various probes have emerged, encompassing carbon nanotubes, quantum dots, organic dyes, and rare-earth nanoparticles [10]. Rare-earth elements exhibit exceptional optical, electrical, and magnetic properties, making them essential raw materials for producing new materials [11]. Due to their narrow emission bandwidth and abundant electronic energy levels, rare-earth elements can be tailored into diverse structures for various applications. Rare-earth nanomaterials used for emission entail doping rare-earth ions into specific nanocrystals. When the excitation wavelength is shorter than the emission wavelength, Stokes emission occurs, and such materials are termed downconversion materials [12], capable of generating NIR-II emission. Unique luminescent properties of rare-earth nanoparticles can be attained through systematic design and optimization. Specifically, rare-earth nanoparticles emitting NIR-II fluorescence possess unique optical and physicochemical properties, such as highly controllable particle size, photostability, long lifespan, and many other technical advantages to promote biomedical applications [13,14]. In cancer imaging, the administrated particles can offer high-quality fluorescence images by passive accumulation in tumor sites, which is widely known as the enhanced permeability and retention (EPR) effect. Utilizing these advantages during surgery can significantly aid surgeons in localizing primary malignancies and metastases and delineate the margin for achieving precise resection [15,16]. To further facilitate the application of NIR-II-guided tumor surgery, novel probes with higher imaging contrast and prolonged tumor retention time are continuously evolving.

In this article, we report a novel downconversion luminescent rare-earth nanoparticle (α-Er NP), which has NIR-II fluorescence characteristics and a high signal-to-background ratio for breast cancer image-guided therapy. After accumulation in the tumor, α-Er NPs demonstrated excellent biocompatibility during both in vitro and in vivo experiments, suggesting potential clinical applications. NIR-II fluorescence bioimaging with α-Er NPs accurately delineated tumor margins, allowing precise removal within a prolonged and stable tumor retention window. Additionally, multimodal imaging, including NIR-II, CT, and ultrasound, facilitated breast tumor detection. Finally, photothermal therapy (PTT) based on NIR-II fluorescence led to a significant reduction in subcutaneous tumor volume. In summary, these findings highlight the excellent fluorescence performance of α-Er NPs, positioning them as a potential probe for real-time imaging and PTT of breast cancer within the NIR-II window.

## 2. Methods

### 2.1. Nanoparticle Preparation and Characteristics

The α-NaErF4@NaYF4 NPs (0.82 g/mL) used in this study were customized by Digi-United Biotechnology (Shanghai, China). The emission spectra were measured using an FLS980 spectrometer from Edinburgh Instruments (Livingston, UK). The morphology and size of the α-NaErF4@NaYF4 nanoparticles were analyzed using transmission electron microscopy (TEM) with an H-600 instrument from Hitachi Ltd. (Tokyo, Japan). Powder X-ray diffraction (XRD) analyses were conducted on a D8 ADVANCE diffractometer from Bruker (Karlsruhe, Germany).

### 2.2. Cell Culture and Cytotoxicity Studies

MCF-7 cells were cultured in RPMI1640 medium with 10% fetal bovine serum. Cellular toxicity was evaluated employing the CCK-8 assay. Cells were seeded at a density of 2000 cells per well in a 96-well plate and incubated for 24 h at 37 °C with 5% CO_2_. The α-NaErF4@NaYF4 NPs were dissolved in RPMI1640 with 10% FBS at concentrations of 0, 2, 4, 8, 16, 32, 62.5, 125, and 250 μg/mL, respectively. Subsequently, cells were exposed to the α-NaErF4@NaYF4 NPs, along with a control medium containing no NPs. After a 24-h incubation period, cells underwent three washes with phosphate-buffered saline and were subsequently incubated in RPMI1640 supplemented with 100 μL and 10 μL CCK-8 solution for 2 h. Optical density was measured at 450 nm, and the results were expressed as a percentage relative to the values of the control wells.

### 2.3. Animal Model of Breast Cancer

Nude mice were maintained in specific pathogen-free (SPF) conditions, with a room temperature ranging from 21 to 25 °C, humidity maintained at 60–70%, and a 12-h alternating light cycle. Food, water, feeding cages, and bedding materials were subjected to sterilization using high pressure. Ethical approval for this experiment was obtained from Shanxi Medical University.

MCF-7 cells of the logarithmic growth stage were collected and washed twice with PBS. Following an additional 3 min of digestion with 0.25% trypsin, the trypsin was aspirated, and the reaction continued for approximately one more minute before terminating the digestion with the addition of medium. Cell suspension was prepared by centrifugation and subjected to two washes with serum-free medium. The cell concentration was standardized to 3 × 10^7^ cells/mL through microscopic counting. Nude mouse skin was sterilized with iodine, and 150 μL of the cell suspension was subcutaneously injected into the right armpit of each mouse. After 10 days of tumor growth, tumor dimensions were measured using vernier calipers, and the tumor diameter was recorded if it exceeded 5 mm. In this experiment, the average tumor volume of our tumor-bearing mice was 150.36 mm^3^.

### 2.4. Ultrasonography and CT Imaging

The Vevo^®^3100 imaging system was employed to capture images of tumor sites in tumor-bearing mice, enabling measurement of both tumor size and location. The IVIS^®^Spectrum CT imaging system was utilized for in vivo CT imaging.

### 2.5. NIR-II Imaging

Tumor-bearing mice received a local injection of α-NaErF4@NaYF4 nanoparticles (NPs) at a dose of 50 μL (0.82 g/mL). Imaging was conducted using an InGaAs SWIR camera (Xenics Cheetah-640CL TE3, Xenics NV, Leuven, Belgium) equipped with an NIR-II light-transmitting lens (Spacecom VF50M SWIR, SPACE, Mitaka City, Tokyo). A filter wheel (Thorlabs, Newton, NJ, USA) with wavelengths ranging from 1000 nm to 1500 nm was affixed to the lens for NIR-II fluorescence capture. Excitation occurred at 808 nm wavelength, with a power of 50 mW/cm^2^ and an exposure time spanning from 30 ms to 1 s. Quantification of fluorescence intensity was performed using ImageJ software (ImageJ 1.53k), and the tumor–background ratio was calculated as the ratio of fluorescence intensity in the tumor to that in the peritumoral tissue.

### 2.6. In Vivo Metabolic Experiments

α-NaErF4@NaYF4 NPs were administered via the tail vein to mice, and NIR-II imaging was conducted to assess its metabolism in vivo. Living mice were intermittently imaged within 48 h post-injection. Tumors were meticulously isolated for ex vivo NIR-II imaging. Following this, fluorescence intensity was quantified using ImageJ for both in vivo and in vitro assessments.

### 2.7. Histopathological Staining

Tumor-bearing mice were euthanized following the intravenous injection of α-NaErF4@NaYF4 NPs at a dosage of 4.8 μL/g body weight. Each organ tissue was initially washed with PBS and subsequently fixed in 4% paraformaldehyde for 12 h. The paraffin-embedded sections were histologically examined by hematoxylin–eosin (HE) staining, and the resulting changes were observed using an upright microscope.

### 2.8. In Vivo Photothermal Therapy

The α-NaErF4@NaYF4 NPs were administered through the local injection of tumor-bearing mice, and PTT was performed 24 h later. Infrared thermal imaging was employed to capture images of the tumor regions. The tumor site of the mice was then irradiated with a near-infrared laser (808 nm, 1.5 W/cm^2^). Infrared thermal imaging was used to record minute-by-minute temperature changes within the tumors, and the tumor diameter was measured post-photothermal treatment. The subsequent temperature change and tumor diameter were recorded. In this experiment, the average tumor volume of our tumor-bearing mice was 83.72 mm^3^. The tumor volumes were calculated using the formula (length × width × width/2).

### 2.9. Statistical Analysis

Quantitative analysis of NIR-II fluorescence images was performed using ImageJ software, and the results were visualized in pseudo color. Experimental data are presented as mean ± standard deviation. Statistical comparisons between groups were conducted using either one-way ANOVA or *t*-test. GraphPad Prism 10.1.0 software was utilized for statistical analysis and graph generation. A significance level of *p* < 0.05 was applied to determine statistical significance.

## 3. Results

### 3.1. Characterization of the α-Er NPs

We characterized α-Er NPs under excitation at 808 nm; near-infrared fluorescence spectra demonstrated maximum emission of the probe within the NIR-IIb range (1300–1700 nm), peaking at 1525 nm (Appendix A). The XRD spectrum revealed that the main size and position of the diffraction peak matched those of the standard Er characteristic diffraction peak (Appendix A). As shown in the TEM images, the α-Er NPs had cubic shapes and superior monodispersibility (Appendix A). In the aqueous solution, dynamic light scattering measured the hydrodynamic particle with an average size of about 24.1 nm. (Appendix A). Collectively, these data indicate the well-executed preparation of α-NaErF4@NaYF4 NPs, characterized by uniform particle dispersion and excellent homogeneity.

### 3.2. NIR-II Imaging of α-Er NPs in Capillary Phantom and Blood Vessels

We utilized an emerging amplified fluorescence at 1525 nm in cubic-phase RENPs with a core–shell nanostructure of α-NaErF4@NaYF4 (Figure 1a). To further evaluate the imaging performance of α-Er NPs and compare the scattering characteristics of photons under different spectra, fluorescence images of capillary tubes containing ICG and α-ErNPs were captured. The optical performance of α-Er NPs in the NIR-IIb region outperforms that of ICG, suggesting a remarkable potential of α-Er NPs in NIR-IIb imaging in vivo (Figure 1b). The vascular images of the mouse abdomen have further shown the advantages of α-Er NIR-IIb imaging. Under 1000 nm observation, the vascular network was hardly recognizable after intravenously administrating α-Er NPs. NIR-II imaging at a longer wavelength (1500 nm) enabled the observation of three blood vessels with greater clarity (Figure 1c). Quantitative analysis of the fluorescence cross-section intensity distribution revealed a significant increase in the imaging contrast ratio as the wavelength increased from 1000 nm to 1500 nm (Figure 1d), resulting in enhanced imaging resolution for recognizing small blood vessels.

### 3.3. The Stability and Biocompatibility of the α-Er NPs In Vitro

α-Er NPs (4.8 μL/g) were administered to mice intravenously. NIR-II scanning, depicted in Figure 2a, revealed a peak signal at 24 h, followed by gradual metabolism. Subsequent ex vivo imaging of organs harvested 24 h post-injection demonstrated heightened signals in the liver, kidneys, and stomach (Figure 2b). HE staining of the organs showed no acute or chronic damage caused by α-Er NPs (Figure 2c). Following a 24-h incubation period of MCF-7 tumor cells with α-Er NPs at various concentrations, cell counting kit-8 (CCK-8) results indicated that the probe did not exhibit significant cytotoxic effects (Figure 3b).

### 3.4. NIR-II/CT/US Imaging and Photothermal Therapy Based on α-Er NPs for Breast Cancer

Nude mice (female, 5 weeks old, n = 6) were inoculated with MCF-7 cells, and the tumor volume reached approximately 150 mm^3^ after 10 days (Figure 3a). Subsequently, this tumor was utilized for NIR-II/CT/US imaging. NIR-II images were obtained 24 h post local injection of α-Er NPs. In the NIR-II imaging scan, the breast tumor site in MCF-7-bearing mice exhibited a higher fluorescence signal (Figure 3b). CT and US (Figure 3c,d) can only provide a suboptimal performance to visualize the margin of breast cancer lesions. In contrast, NIR-II demonstrated a superior imaging resolution. Encouraged by the favorable outcomes observed in vivo with NIR-II imaging, we proceeded to assess the photothermal antitumor capability and efficacy of the α-Er NPs in MCF-7 tumor-bearing mice. Throughout the photothermal treatment process, the temperature variation within the tumor was monitored every minute using a thermal imaging device. As the irradiation time progressed, the temperature at the tumor sites gradually elevated (Figure 3e). Following a 5-min irradiation period, the tumor temperature dramatically surged to above 50 °C (Figure 3f), a temperature deemed sufficiently high for the effective ablation of cancer cells. Meanwhile, to evaluate the antitumor effect of PTT using α-Er NPs, the tumor volume was measured and recorded every day. After performing photothermal therapy on days 0, 4, 8, and 12, images of isolated mouse tumors were obtained (Figure 3g). The tumors exhibited a significant reduction in size following photoirradiation at 808 nm (Figure 3h). Figure 3h illustrates the tumor volume ratio of mice at different treatment times. Our findings affirm that near-infrared light-treated breast tumors exhibit significant tumor suppression, underscoring the potential of RENPs-mediated photothermal therapy for tumor ablation.

## 4. Discussion

Breast cancer continues to be the primary cause of mortality among women globally. Despite a significant reduction in breast cancer mortality rates in recent years due to increased awareness of early diagnosis and the application of hormone therapy, the occurrence of breast cancer metastasis still dramatically decreases the five-year survival rate for patients. Therefore, achieving early and accurate diagnosis, as well as complete removal of breast cancer, is crucial [17,18].

Numerous studies have underscored the efficiency of conventional imaging techniques for breast cancer diagnosis, including magnetic resonance imaging (MRI), ultrasound (US), and computed tomography (CT) [19,20,21]. Nevertheless, integrating these imaging modalities into operating rooms faces great challenges, and notably, they lack the capability to offer real-time feedback to surgeons reliably [22]. Despite the utilization of intraoperative ultrasound for certain cancer surgeries, depending on a single imaging modality often proves inadequate for distinguishing between cancerous and non-cancerous regions [23]. NIR-II imaging, as an emerging optical imaging modality in recent years, can afford excellent sensitivity, resolution, and real-time capability to visualize cancer lesions intraoperatively. In comparison to the visible and NIR-I regions, NIR-II imaging exhibits smaller tissue absorption and scattering and lower biological autofluorescence, resulting in a higher signal-to-background ratio and superior resolution [24,25,26,27].

Since 2020, NIR-II image-guided surgery has been applied in clinics; the impressive outperformance strongly facilitates the research and application of NIR-II intraoperative imaging [28,29]. Although ICG offers a convenient approach to perform NIR-II imaging in various cancers, such as breast cancer, gynecological cancer [30,31,32,33], and digestive tract cancer [34], the unstable tumor retention might affect surgeons to accurately locate and resect tumors [33,35]. In order to address the diverse and multi-level clinical demands, taking into account the advantages of NIR-II imaging, the development of novel NIR-II optical probes for in vivo imaging represents an optimal choice.

Lately, various NIR-II fluorescence probes, including RENPs [13,14], quantum dots [36,37], single-walled carbon nanotubes [9,38], and organic molecules [39], have undergone significant advancements. Among these probes, RENPs exhibit significant advantages, including large Stokes shifts, narrow-peak and multi-peak emission spectra, and optical stability, making them well-suited for biomedical applications [40,41]. Yang et al. reported the liposome-coated lanthanide nanoparticles for fluorescence imaging of brown adipose tissue and intraoperative multi-spectral imaging of intestinal vessels in the NIR-II [41]. Liu et al. synthesized novel up/downconversion luminescent RENPs for on-demand gas therapy of glioblastoma guided by NIR-II fluorescence imaging [42]. Additionally, Wang et al. developed dye-sensitized RENPs for dynamic imaging of vascular network-related diseases [43]. These findings underscore the broad potential of RENPs in the field of bioimaging.

This study successfully utilized a biocompatible and excretable NIR-II rare-earth nanomaterial, α-Er NPs, for NIR-II fluorescence imaging. The rare-earth probe enabled non-invasive fluorescence-guided tumor detection and in vivo high spatial resolution sensing, accurately delineating tumor margins and facilitating precise tumor excision to prevent recurrences. Moreover, NIR-induced photothermal therapy shows promise as a non-invasive and non-destructive alternative to traditional methods for treating tumors, with effective photothermal therapy heavily dependent on visualization imaging techniques. Photothermal therapy operates by employing photothermal nanoparticles (such as gold [44] or graphene [45]) to absorb light energy, subsequently converting it into heat energy upon exposure to light irradiation. When the photothermal effect acts on tumor cells, it induces phenomena such as cell dissolution and enzyme release, resulting in cell necrosis and protein denaturation, thereby achieving the therapeutic effect on tumors [46]. This paper primarily discusses the application of NIR-induced rare-earth nanoparticles for imaging-guided photothermal therapy in breast cancer.

In addition, hyperthermia was induced, and effective tumor ablation was achieved under 808 nm laser irradiation. Additionally, in vitro and in vivo experiments confirmed the biocompatibility of α-Er NPs, with the majority being eliminated from the body through the hepatic clearance pathway within 72 h. Our observation revealed that α-Er NPs possess characteristics such as ultra-long wavelength fluorescence, rapid metabolism, and precise labeling of tumor-positive margins. Meanwhile, multi-modal imaging combining CT and US can further provide comprehensive tumor information. These features confer significant clinical potential to achieve accurate identification and photothermal treatment of breast cancer.

In summary, our research results demonstrate intraoperative tumor-specific NIR-II fluorescence imaging utilizing α-Er NPs with long wavelengths and optical stability, offering potential benefits in the identification of breast cancer during surgery. This approach does not impose unnecessary interference with standard surgical procedures. The integration of optical imaging techniques with tumor diagnosis and treatment strategies has the potential to transform the paradigm of tumor surgery, providing unique opportunities for intraoperative detection and phototherapy of malignancies and metastases.

## 5. Conclusions

Breast cancer emerges as a primary contributor to the incidence, disability, and mortality experienced by women, with its prevalence persistently increasing. Given the challenge posed by vague tumor boundaries, the incomplete clearance of tumors significantly impacts patient prognosis. NIR-II imaging, acknowledged for its exceptional imaging capabilities, represents a distinctive and innovative approach to the diagnosis and treatment of breast cancer. For individuals with breast cancer, high-resolution imaging under NIR-II guidance facilitates the precise visualization of tumor margins, enhancing accurate surgical navigation. Moreover, the integration of NIR-II imaging with rare-earth nanoparticles exploits their efficient photothermal conversion effect, ensuring targeted and effective eradication of tumor cells. This amalgamation introduces a secure and effective novel strategy for the integration of breast cancer diagnosis and treatment, offering substantial promise for clinical applications.

## Figures and Tables

**Figure 1 biology-13-00156-f001:**
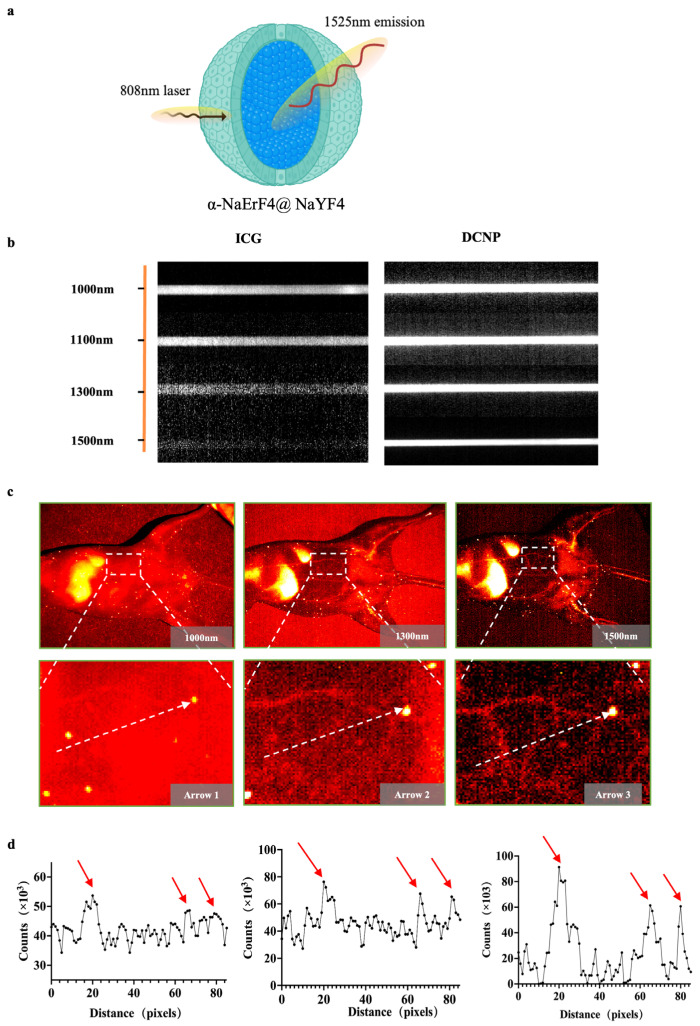
α-Er NPs NIR-II imaging of blood vessels and capillary phantom. (**a**) Schematic of the core–shell structured α-Er nanoparticles. (**b**) Intensity images of capillaries containing ICG and α-Er NPs. (**c**) NIR-II image of mouse abdominal vessels with different NIR-II long-pass filters on an InGaAs camera. (**d**) Fluorescence cross-sectional intensity distribution of abdominal vessels in the NIR-II window (white dotted line in (**c**)), and the peak pointed by the red arrow in the curve is the location of the blood vessel.

**Figure 2 biology-13-00156-f002:**
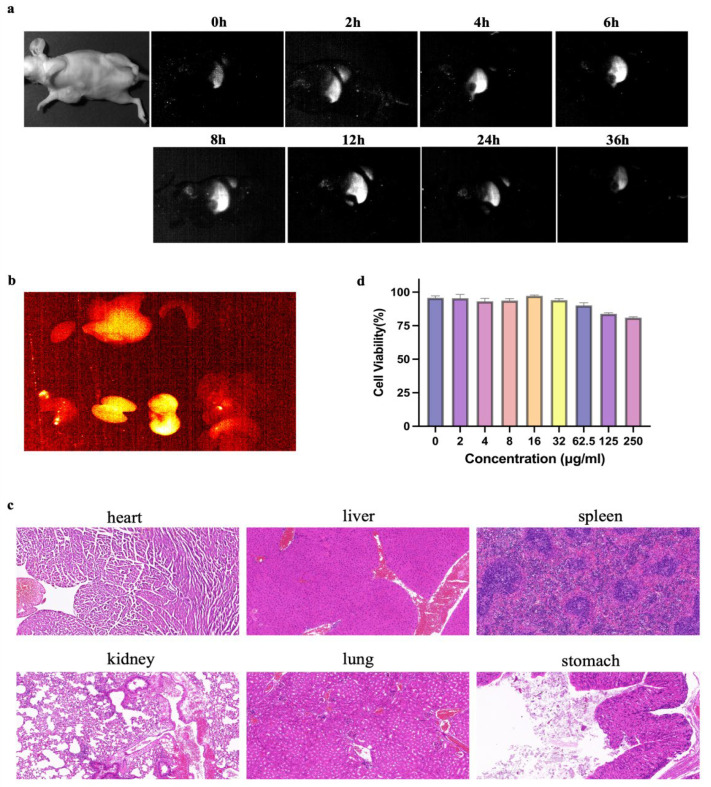
Pharmacokinetics and cytotoxicity assay of α-Er NPs. (**a**) Representative examples of whole-body imaging at different time points of intravenous injection of α-Er NPs. (**b**) Fluorescence images of α-Er NPs in epidermal tumor of the nude mice and the corresponding harvest organs (left to right and to bottom: heart, liver, spleen, lung, kidney, stomach, and intestine). (**c**) Representative H&E pathological sections of vital organs, including heart, liver, spleen, lung, kidney, lung, and stomach harvested from the mice after being administered with α-Er NPs post-injection. No abnormal lesion or injury was observed in the histological examination of these organs. Scale bar: 50 μm. (**d**) CCK-8 assay of MCF-7 cells demonstrated no obvious cytotoxicity after 24 h of incubation with diverse concentrations of the α-Er NPs. The concentration gradients were 0, 2, 4, 8, 16, 32, 62.5, 125, and 250 μg/mL, respectively.

**Figure 3 biology-13-00156-f003:**
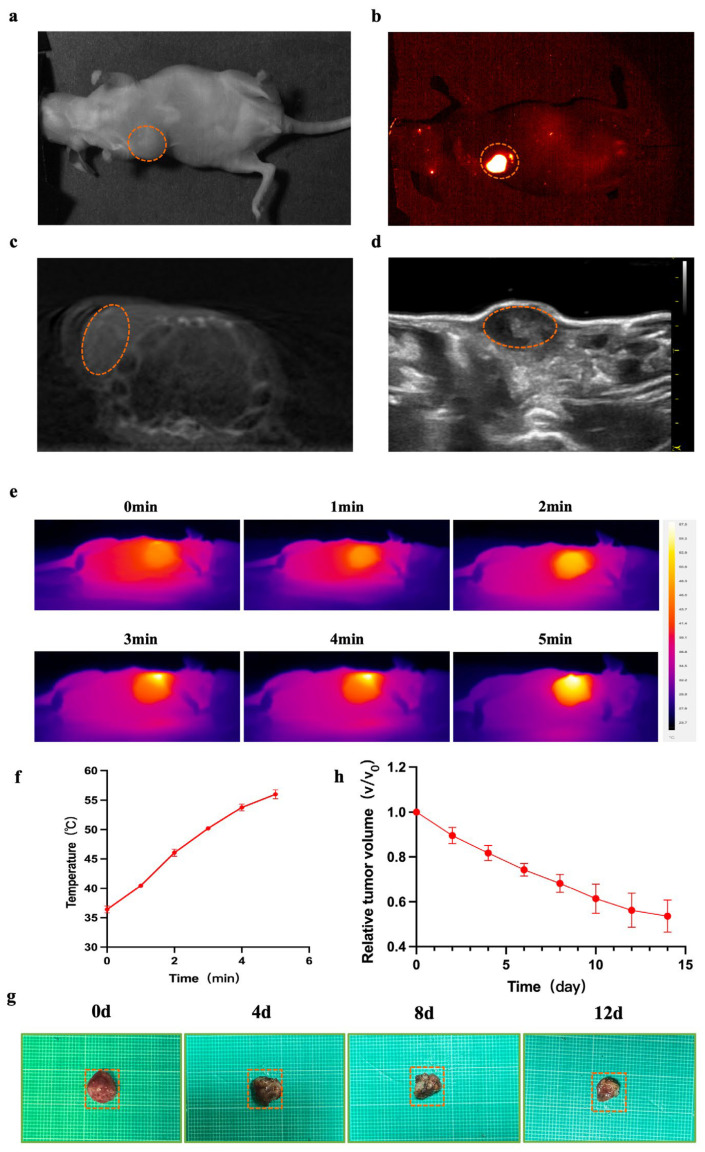
NIR-II/CT/US imaging and PTT for tumor. (**a**–**d**) White light and NIR-II/CT/US images of α-Er NPs were captured (Dotted orange circles indicate tumor). (**e**) Thermal images of mice and the (**f**) corresponding tumor temperature changes under an 808 nm laser irradiation (1.5 W cm^−2^) for 5 min after the local injection of the agent for 24 h. (**g**) The picture shows tumors extracted from the different treatment times. (**h**) The tumor volume ratio of mice at different treatment times.

## Data Availability

The raw data supporting the conclusions of this article will be made available by the authors, without undue reservation, to any qualified researcher.

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
