# Peer review of "The Down-Shifting Luminescence of Rare-Earth Nanoparticles for Multimodal Imaging and Photothermal Therapy of Breast Cancer"

_biology, 2024, doi:10.3390/biology13030156_

Round 1
Reviewer 1 Report
Comments and Suggestions for Authors
The paper is interesting, presents orginal contributions. It is well written and well organized. Mandatory changes before to be published should be focused on the literature review of photothermal applications of nanoparticles. For example, I strongly suggest to cite the paper Exploiting gold nanoparticles for diagnosis and cancerc treatments, Nanotehnology, 2021, vol. 32 192001, doi: 10.1088/1361-6528/abe1ed
Author Response
Many thanks to you for your careful review and a series of sincere and useful suggestions for our manuscript. Here is a point-to-point response for all your concerns and we hope that we have addressed all of your concerns.
Changes within the manuscript we will identify with a yellow background color.

Reviewer 2 Report
Comments and Suggestions for Authors
The manuscript presents an intriguing application of rare-earth nanoparticles that emit in the near-infrared-II window (NIR-II), specifically NaErF4@NaYF4, that have suitable biocompatibility. The in vivo imaging results demonstrate the nanoprobe's imaging capability and thermal conversion efficiency in the NIR-II range under 808 nm laser irradiation. It is worth noting that integrating NIR-II imaging technology for imaging and treatment provides interesting opportunities in biomedical fields, and this manuscript explores this new system. However, there are several issues that need to be addressed, mainly related to a lack of information about the mechanism of intake (no comment on pharmacokinetics and speculation about intracellular fate). Additionally, the toxicity aspect is given little attention, which is of paramount importance for translating this idea into clinical settings. The thermal properties are only reported macroscopically, with no information regarding the material's thermal properties with quantification in test tubes and cells. It is also unclear how much the heat released by the nanoparticles contributes to the direct illumination at this fluence.
Comments on the Quality of English LanguageThe manuscript requires only limited editing.
Author Response

(The authors gave the same response as above.)

Reviewer 3 Report
Comments and Suggestions for Authors
The authors explain the importance of innovative nanotherapeutics to treatment of breast cancer and, in this case, a multi-functional nanoparticle capable of produce photothermal effect combined to CT and ultrasound imaging. It is also highlighted the use of fluorescence molecular imaging for cancer treatment and the differences between the two near-infrared windows. Regarding the nanoparticles chosen the rare-earth nanoparticles, it is clear the many advantages that they present to use for cancer treatment. The authors present these innovative nanoparticles benefits to the breast cancer treatment including the initial characterization of the nanoparticles and a broad efficacy study in vivo.
1 – I would suggest a brief explanation in the introduction about the materials that can be used for these rare-earth nanoparticles as maybe not intuitive for all readers what kind of material these nanoparticles are related.
2 – In the methods section of the nanoparticles application in the mice model (2.5 and 2.7) would be good to express the dose in grams or milligrams related to the weight of the animals.
3 – In the image 1-d, if possible increase the size of the title of the axis x and y because it is not possible to read and it seems distance is written with "s".
4 – It was not very clear for me the meaning of the figure 3 g-h. Are these data obtained after PTT? Because I think it is expected to reduce the size and not increase. I understand by image 3-g that volume decreased but the graph 3-h is not clear, it is very confusing because on the caption it is not explained what v/v0 means. Even considering that is volume on day “x” divided by initial tumor volume, it is not very clear the graph representation. I think the representation should show a decrease tendency and it is not what it is observed and may lead to wrong conclusions. Maybe reconsider how to express the tumor volume information because I think it is a very nice result from the experiments and the quantitative evolution should be as clear as the qualitative (photos shown in figure 3-g).
Author Response

(The authors gave the same response as above.)

Round 2
Reviewer 1 Report
Comments and Suggestions for Authors
The authors addressed all the issues evidenced after first revision